# New 3D Printing Strategy for Structured Carbon Devices Fabrication

**Gabriel Delgado-Martín** [1], **Nicolás Rodríguez** [2,†], **María Isabel Domínguez** [1,*], **Yazmin Yaneth Agámez** [2], **Marcela Martínez Tejada** [1], **Estela Ruíz-López** [1], **Svetlana Ivanova** [1] and **Miguel Ángel Centeno** [1]

[1] Departamento de Química Inorgánica e Instituto de Ciencia de Materiales de Sevilla, Centro Mixto CSIC—Universidad de Sevilla, 41092 Sevilla, Spain; gabriel.delgado@icmse.csic.es (G.D.-M.); leidy@us.es (M.M.T.); estela.ruiz@icmse.csic.es (E.R.-L.); sivanova@us.es (S.I.); centeno@icmse.csic.es (M.Á.C.)

[2] Departamento de Química, Facultad de Ciencias, Universidad Nacional de Colombia, AK 30 No 45-03, Bogotá 111321, Colombia; nrodriguezri@unal.edu.co (N.R.); yyagamezp@unal.edu.co (Y.Y.A.)

[*] Correspondence: mdominguez1@us.es

[†] Current address: Departamento de Química, Facultad de Ciencias, Universidad de la Amazonia, Calle 17 Diagonal 17 con Carrera 3F, Florencia 180001, Colombia.

**Abstract:** This work shows a new method for the preparation of 100% carbon-structured devices. The method is based on resorcinol-formaldehyde polymerization, using starch as a binder with the addition of a certain amount of external carbon source before polymerization. Molds obtained by 3D printing are used to shape the structured devices in the desired shape, and the ultimate pyrolysis step consolidates and produces the carbonaceous devices. The proposed method allows obtaining supports with different textural and surface properties varying the carbonaceous source, the solvent, or the pyrolysis conditions, among other factors. The as-obtained devices have demonstrated their usefulness as palladium supports for the gas-phase formic acid dehydrogenation reaction. The monolith shows a high conversion of formic acid (81% according to $H_2$ production) and a high selectivity towards hydrogen production at mild temperatures (80% at 423 K).

**Keywords:** microreactors; 3D print; carbon microstructured devices; $H_2$ production; formic acid decomposition

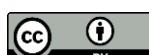

## 1. Introduction

The use of structured and micro-structured catalysts has grown exponentially in recent years as being one of the most efficient solutions to decrease the operational problems and costs related to the mass and heat transport limitations in many industrial reactions. These devices allow high space velocities resulting in process intensification, delocalization and demand-to-manufacture production, continuous processes with higher safety, etc.

As outstanding characteristics of the structured catalysts and monolithic porous structures in particular, we can enumerate (i) the high void fraction and large geometrical surface area [1], providing large contact surfaces between catalyst and reagents, (ii) the low-pressure drops under flow conditions and (iii) the high mechanical stability and dust tolerance. The application of such devices in automobile exhaust (three-way catalyst) [2,3] and NOx emissions removal [4] is well-known and widespread. However, in the real chemical conversion industry, the application of structured catalysts started becoming an innovative topic [5] in the last few years.

The great number of possible catalytic applications requires adapting the nature, morphology, size, and properties of the structured system. Bulk monoliths, monoliths

with parallel channels of different geometry (square, circular, etc.), open cell foams, periodic cellular architectures, and monoliths with different shapes (square, cylindrical etc.) must adapt to the fluid dynamics of every particular process.

Of special importance is the nature of the substrate forming the structure since it must be selected according to the reaction working conditions: temperature, reaction atmosphere, gas or liquid media, etc. The substrates can be metallic, glass/vitreous, ceramic, or based on carbon materials, the latter being the subject of this study.

Although considered unstable in the presence of hydrogen at temperatures above 700 K and in oxygen above 500 K [6], carbon substrates present several advantages, such as (i) high stability in both acidic and alkaline media [7], (ii) chemical inertness [8]; (iii) ease of active phase recovering; (iv) low cost [9], and (v) tuneable surface chemistry and porosity properties [8]. It is possible to modify the substrate's hydrophobic/hydrophilic character, i.e., the chemical nature of the surface and its porous structure, by changing the carbon precursor used during the structure reactors preparation and activation process [6].

Up to now, the fabrication of carbon-based substrates has been essentially based on extrusion after mixing with a certain amount of non-carbonaceous compounds, such as metals. Only in recent years have new strategies for pure carbon monolith production been reported, although their number remains rather scarce. The advances in the production and application of bulk carbon monoliths [10,11], cell foams [12], or parallel channel monoliths [13–15] are summarized in Table 1.

**Table 1.** Preparation methods and applications of different monolithic carbon structures.

| Material | Preparation | Application | Reference |
|---|---|---|---|
| (Bulk) monolithic carbon aerogels | Sol-gel polymerization of phenolic resin within melamine foam (MF), ambient pressure drying, and co-carbonization. | Temperature thermal insulators Organics absorption | [10] |
| (Bulk) monolithic carbon xerogels | Sol-gel polymerization of resorcinol and formaldehyde in water, $CsCO_3$ as the polymerization catalyst. | Crude oil removal from oil-saltwater emulsions | [11] |
| (Nickel-nitrogen-doped) carbon foam | After curation, the organic gel is immersed in acetone, microwave drying, and carbonization in a nitrogen flow at 900 °C. | Electrode for $CO_2$ electroreduction | [12] |
| Monolithic biochar | Impregnation of a polyurethane foam template with a solution of coal tar pitch in tetrahydrofuran. Drying at room temperature, oxidation in air at 350 °C, and carbonization at 950 °C in argon. | Catalytic conversion of toluene | [13] |
| Monolithic cobalt-nitrogen-carbon frameworks | Hydrothermal treatment of wood cylinders in Co-metal solution (200 °C). Drying at 105 °C and pyrolysis at 800 °C in an $N_2$ atmosphere. | Electrochemical synthesis of hydrogen peroxide in acidic media | [14] |
| Hierarchical monolithic carbon | Wood carbonization at 1000 °C in an $N_2$ atmosphere. | Hydrogen evolution reaction | [15] |

Among the various synthetic routes, carbon gels are preferred because of their high purity, homogeneity, and properties tune possibility (morphology, porosity, and surface chemistry). Their versatility is based on the possibility of adapting the precursor concentrations and gel synthesis parameters to obtain materials with targeted properties for a given application [16]. For example, the pH of the precursor solution governs the size of

the final meso/macropores [17], but their distribution in the final carbon xerogels is controlled by combining different synthesis conditions and chemical activation with KOH [18].

Once prepared, the carbon precursors must be consolidated in carbon monoliths of sufficient mechanical strength for possible future industrial applications. This strength is highly dependent on the starting carbon material and preparation process. Compressive strength of 10.1 MPa has been described for a monolith with 9 channels/cm² and a wall thickness of 900 μm. An increase of the channels to 62 channels/cm² and a decrease in the wall thickness to 165 μm resulted in lower compressive strength of 4.4 MPa per monolith [8]. Carbon monoliths produced from activated carbon powder are mechanically weak [19], while the use of phenol-formaldehyde resins results in integral carbon structures with superior mechanical properties [20]. These resins are also important for the 3D printing strategy of preparation of carbon-structured substrates recently patented by the authors of this study [21]. This method allows the preparation of integral three-dimensional carbon structures using carbonaceous materials of a very different nature. The novelty of the fabrication process resides in the use of resorcinol-formaldehyde polymer mixed with a natural polymeric binder such as starch [22] and a carbonaceous source of different natures to obtain carbon monoliths with fully defined geometries and retaining the properties of the used carbon source. The process consists of five steps: (i) preparation and stabilization of organic solution precursor for resorcinol-formaldehyde polymerization; (ii) grinding and sieving of the additional carbonaceous source; (iii) mixing of the latter with the organic solution and (iv) packing in a 3D printed mold and (v) polymer curing and pyrolysis to obtain the final carbon monolith [21].

During the process of fabrication, various parameters can be modified, e.g., the nature of solvent or aldehyde in the polymerization step, the nature and quantity of additional catalysts or binders, the nature of the added carbon (commercial activated carbon, charcoal, biochar or nanotubes, etc.) or the organic solution/carbon (mL/g) ratio. The same is available for the curing and pyrolysis conditions. Parameters such as time, temperature and type of vessel during the curation process or atmosphere, temperature, physical/chemical activation, and heating rate of pyrolysis, and also treatments for monolith demolding can be adjusted. All these factors affect the characteristics of the finally obtained structured carbon materials in terms of mechanical, textural, and surface properties. In previous work, Rodríguez et al. [22] studied the influence of starch addition as a binder in the resulting carbon xero- and aerogels by resorcinol-formaldehyde polycondensation. They found that starch allowed carbon xerogels and aerogels surface and textural properties modulation, achieving structural properties similar to those of aerogels with a faster and less expensive process of fabrication (convective vs. supercritical drying).

In this line, the present work devotes to the study of the influence of the solvent nature in the polymerization stage, the heating rate in the pyrolysis stage, and the initial carbon source on the final properties of structured carbonaceous devices. The utility of the prepared, structured systems is tested in the gas phase hydrogen production via formic acid dehydrogenation using a low charge of Pd metal as the active site.

## 2. Results and Discussion

As indicated in the introduction, the carbon monoliths are prepared in a 5-step process using pre-fabricated polylactic acid (PLA) molds that result in carbon devices exemplified in Figure 1. A detailed description of the preparation procedure is described in Section 3.

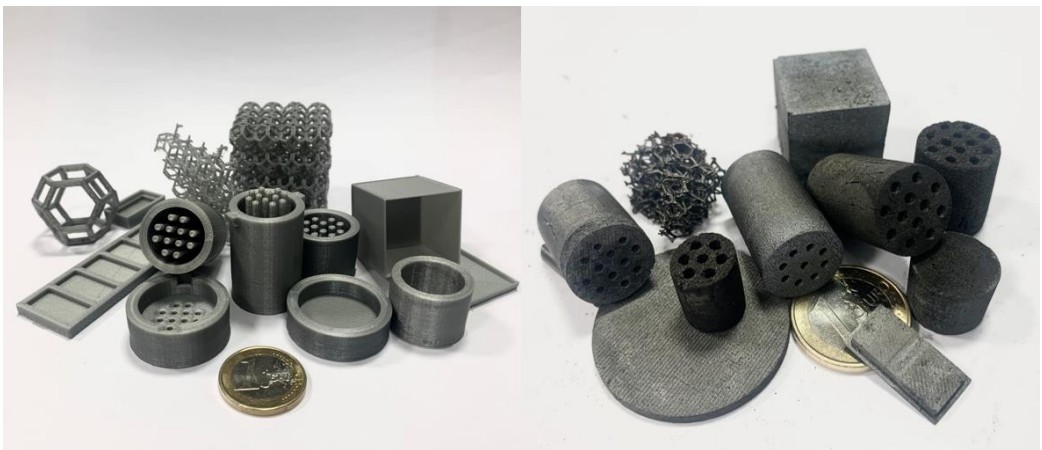

**Figure 1.** Examples of prepared PLA molds (**left**) and carbon monolithic devices (**right**).

As shown in Figure 1, the method of preparation allows the fabrication of pieces with different sizes, morphology, and number of channels. The final quality of the carbon devices can be influenced at every step of the process. Herein only the nature of the solvent used in the polymerization stage and the effect of the heating rate in the pyrolysis stage will be discussed.

### 2.1. Influence of the Nature of Solvent in the Polymerization Stage on the Carbon Structured Device

Different solvents have been tested (water, 96% ethanol, polyethylene glycol (PEG), or PEG + 1% polyvinyl alcohol (PVA). The rest of the variables of the process and added carbonaceous source (activated charcoal Darco Sigma-Aldrich) are kept constant.

The monoliths have been prepared, as explained in Section 3, using PVA molds. The final carbon monolith measurements are 1.5 cm in diameter, 1 cm high, and 28 entire channels, resulting in a device with 21.5 channels/cm$^2$ (Figure 2). The nomenclature of the obtained monoliths is resumed in Table 2.

**Table 2.** Nomenclature of the monoliths prepared with different solvents.

| Solvent | Nomenclature |
| --- | --- |
| Water | D_H$_2$O |
| Ethanol | D_EtOH |
| Polyethylene glycol (PEG) | D_PEG |
| PEG + 1% Polyvinyl alcohol (PVA)l | D_PVA |

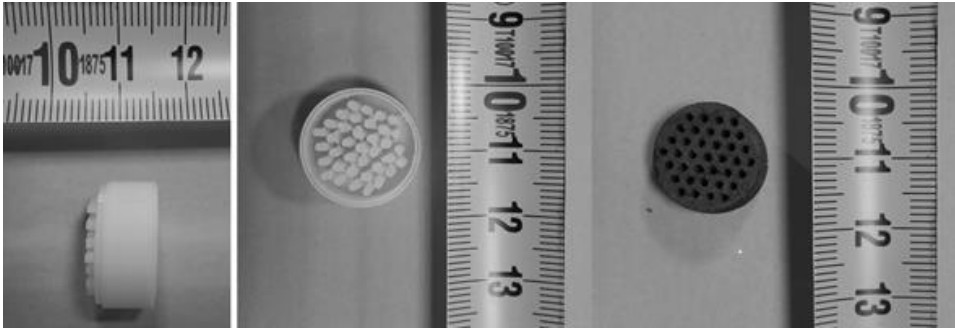

**Figure 2.** PVA mold (left) and monolithic carbon device (right) [21].

Figure 3 presents the diffractograms obtained for the commercial activated charcoal and the monoliths made with different solvents. They all exhibit the characteristic features of amorphous carbon materials: broad signals at 25° and 44° 2θ attributed to the (002) and

(100) planes, respectively. The first one is due to aromatic ring ordering and the second one is related to the degree of aromatic ring condensation. The sharp peaks located at 21°, 23°, 27°, 36°, and 51° 2θ correspond to inorganic phases (quartz and cristobalite $SiO_2$) present in the commercial activated carbon [23].

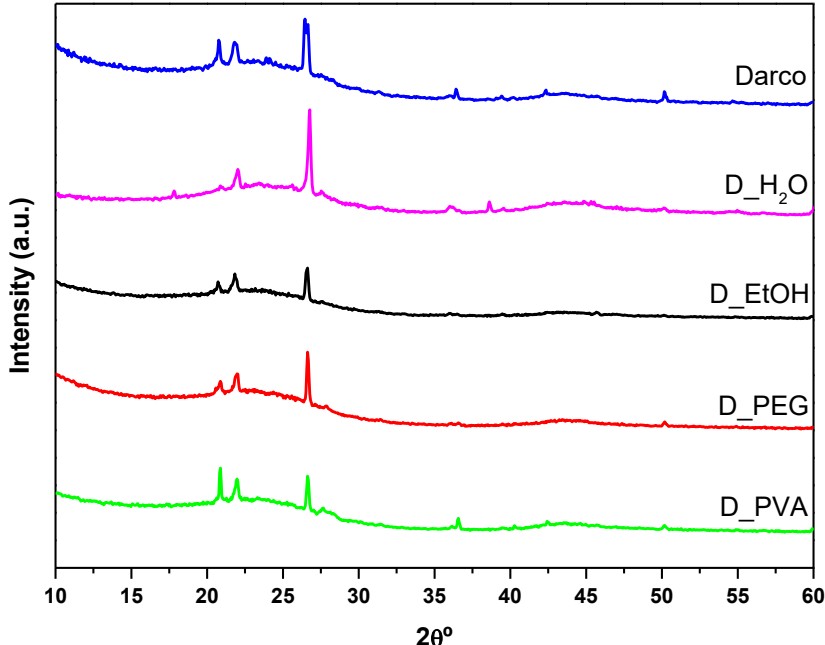

**Figure 3.** The X-ray diffraction patterns for the commercial activated charcoal Darco and the monoliths made with different solvents.

Figure 4 shows the isotherms obtained for the commercial activated carbon and the four monoliths. It should be noted that, in order to obtain the adsorption-desorption isotherms, it was necessary to crush and sieve the material to a homogeneous particle size of less than 500 μm. All the carbons present type IV isotherms according to the IUPAC classification, denoting the presence of mesopores [24]. However, Darco and D_H₂O isotherms are slightly different and can be classified as a combination of type I and IV isotherms suggesting the coexistence of micro and mesopores. The hysteresis cycles for commercial activated charcoal (Darco), and D_H₂O can be classified as type H4, typical of activated carbons with flexible slit-type pores. On the other hand, an important change is evidenced in the isotherms of D_PVA, D_EtOH, and D_PEG with respect to the hysteresis cycles, classified as type H2 and corresponding to pores formed by parallel lamellae. This type of hysteresis is characteristic of carbonaceous materials with a high proportion of graphite in their structure [24].

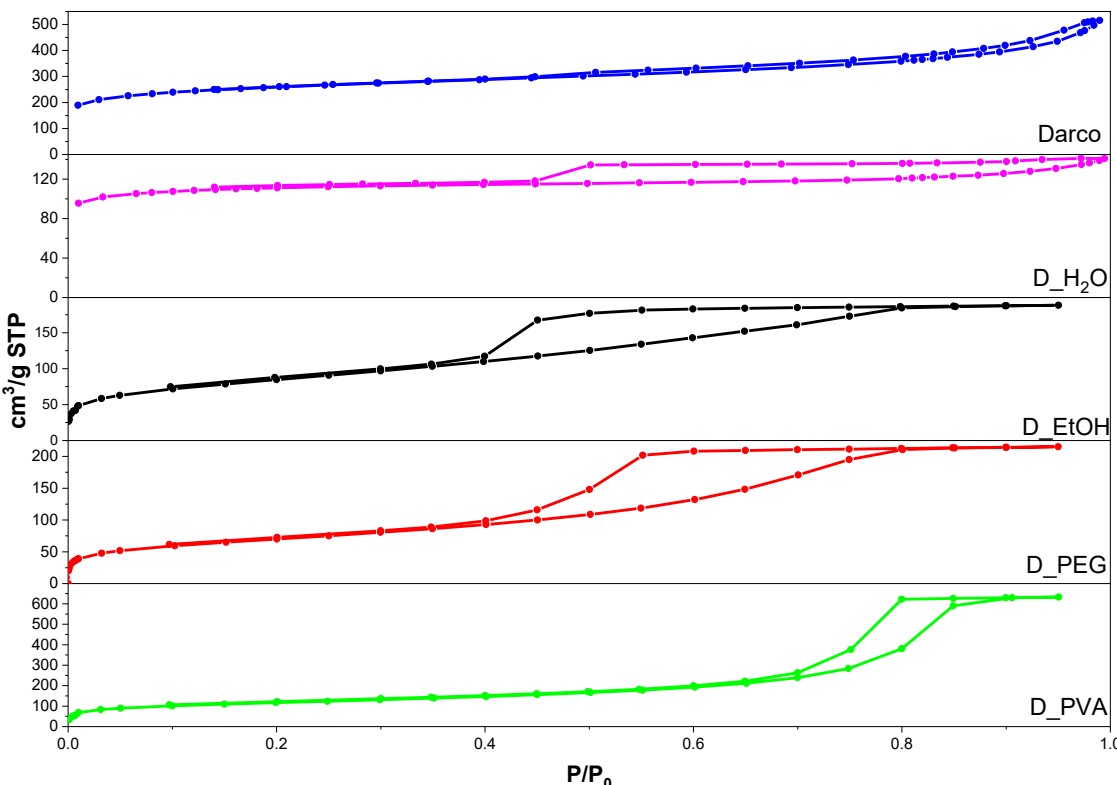

**Figure 4.** $N_2$ adsorption and desorption isotherms at 77 K for the carbonaceous materials.

A summary of the structural parameters obtained from X-ray diffraction and the textural properties of the monoliths prepared using different solvents and compared to the properties of the starting activated carbon is shown in Table 3. The value of the interlayer spacing ($d_{002}$) of the materials remains constant. Only in D_EtOH is a significant change in the average crystallite height (along the c axis, Lc = 1.8 nm) observed in comparison to the values calculated for commercial activated carbon and the other monoliths. The average crystallite width (along the a axis, La) values show either reduction with respect to the commercial activated carbon for the D_EtOH monolith or an increment for the D_PVA monolith.

**Table 3.** Textural properties and structural parameters obtained for commercial activated carbon monoliths varying the type of solvent.

| Sample | $S_{BET}$ (m²/g) | V mp (cm³/g) | S mp (m²/g) | S ext (m²/g) | mp % | APD DFT (nm) | Vp (cm³/g) | $d_{002}$ (nm) | Lc (nm) | La (nm) |
|--------|------|------|------|------|------|------|------|------|------|------|
| AC Darco | 892 | 0.215 | 470 | 422 | 52.7 | 0.8 | 0.73 | 0.37 | 1.2 | 4.4 |
| D_H₂O | 375 | 0.138 | 297 | 78 | 79.2 | 0.7 | 0.21 | 0.37 | 1.2 | 3.1 |
| D_EtOH | 301 | 0.078 | 171 | 130 | 56.8 | 0.5 | 0.40 | 0.37 | 1.8 | 2.9 |
| D_PEG | 213 | 0.065 | 132 | 81 | 62.0 | 0.7 | 0.24 | 0.37 | 1.4 | 4.4 |
| D_PVA | 381 | 0.103 | 223 | 166 | 58.5 | 0.5 | 0.39 | 0.37 | 1.2 | 5.8 |

mp: micropore; ext: external; APD$_{DFT}$: average pore diameter by density functional theory.

In terms of specific surface area, all the monoliths show lower values than the starting Darco and a higher percentage of microporosity, 62 or 79% for the monoliths prepared with PEG and water, respectively. As indicated above, adding the starch as a binder increases the microporosity of the obtained carbon and is related to the polymeric structure formed by starch during the gelation step and remaining after the pyrolysis [22,25]. The

APD$_{DFT}$ values calculated with a non-local function show the presence of micropores ranging from 0.5 to 0.8 nm for all solids. From these results, it can be deduced that in all materials, both the microporous and mesoporous fractions contribute to the specific surface area (S$_{BET}$).

Thus, it is possible to obtain monoliths with completely defined pore volumes, surface areas, and crystallite sizes and different from the starting carbonaceous material only by using different solvents. The possibility of using different solvents provides the advantage of being able to work with carbonaceous materials with hydrophilic, hydrophobic, or amphoteric characteristics for the production of monoliths.

### 2.2. Influence of the Heating Rate in Pyrolysis Step on the Carbon Structured Device

Different heating rates were studied by preparing pellet-type monoliths using water as a solvent and carbon xerogel (CXG) as a carbon source. The heating rate used ranges from 2 to 20 K/min, with all other variables constant. The full methodology for monolith preparation is given in Section 3. The obtained monoliths have been named according to the heating rate, as shown in Table 4.

**Table 4.** Nomenclature of monoliths prepared with different carbonization heating rates.

| Heating Rate (K/min) | Nomenclature |
|:---:|:---:|
| 2 | 2 K/min |
| 5 | 5 K/min |
| 10 | 10 K/min |
| 15 | 15 K/min |
| 20 | 20 K/min |

Figure 5 shows the diffractograms of the pellet-type monoliths obtained by varying the heating rate in the pyrolysis stage. All the diffractograms are typical of amorphous carbonaceous materials with characteristic peaks corresponding to the (002) and (100) planes. The total absence of any other type of signal is noted, thus indicating the production of all-organic integral monoliths with high carbon content [26].

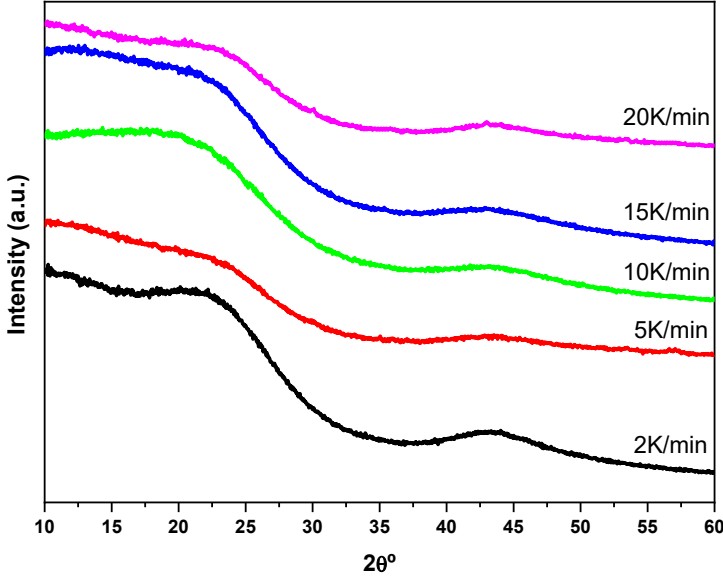

**Figure 5.** The X-ray diffraction patterns for pellet-type monoliths obtained at different heating rates.

Table 5 shows the structural parameters obtained for the monoliths compared to those of the xerogel used as a carbon source. The interlayer spacing ($d_{002}$) oscillates around the same value, 0.38 nm. The crystallite height (Lc) shows values between 1.3 and 1.5 nm, higher than those of the xerogel, and the crystallite size varies between 2.4 and 3.1 nm, with the xerogel showing an intermediate value. Although no clear relationship between the heating rate and these parameters is found, the largest crystallites are obtained using the highest heating rate.

**Table 5.** Structural and textural parameters calculated for pellet-type monoliths obtained by varying the heating rate in the pyrolysis stage and for xerogel used as a carbonaceous source.

| Sample | $S_{BET}$ (m²/g) | V mp (cm³/g) | S mp (m²/g) | S ext (m²/g) | mp % | Vp (cm³/g) | $d_{002}$ (nm) | Lc (nm) | La (nm) |
|---|---|---|---|---|---|---|---|---|---|
| CXG | 596 | 0.12 | 271 | 325 | 45 | 0.59 | 0.4 | 1.1 | 2.8 |
| 2 K/min | 524 | 0.16 | 257 | 266 | 49 | 0.59 | 0.38 | 1.3 | 2.5 |
| 5 K/min | 560 | 0.15 | 430 | 130 | 77 | 1.35 | 0.38 | 1.5 | 2.7 |
| 10 K/min | 590 | 0.13 | 289 | 301 | 49 | 1.14 | 0.40 | 1.3 | 2.7 |
| 15 K/min | 558 | 0.12 | 300 | 258 | 54 | 0.66 | 0.39 | 1.5 | 2.4 |
| 20 K/min | 432 | 0.12 | 309 | 123 | 72 | 0.67 | 0.38 | 1.5 | 3.1 |

mp: micropore; ext: external.

Figure 6 shows the isotherms of the monoliths obtained at different heating rates. According to IUPAC, all isotherms present a characteristic type IV and I shape, indicating mesoporous materials with high microporosity contribution confirmed by the adsorbed volumes at very low relative pressures. All hysteresis cycles have similar shapes and correspond to type H4 hysteresis, associated with flexible slit-like pores [24].

A summary of the textural properties of the monoliths obtained at different heating rates during the pyrolysis stage is given in Table 5 compared to those of the xerogel used as a carbonaceous source. The BET surface area values vary between 432 and 590 m²/g. The micropore volume presents values between 0.12 and 0.16 cm³/g, and it is observed how this volume decreases with increasing the heating rate. The micropore area percentages of the monolith remain between 50 and 70%, being the manufactured monolith with a heating rate of 5 K/min being the one with the highest microporosity percentage. The external area, mainly due to the existence of mesopores and macropores in the monoliths, varies between 130 and 301 m²/g, and the pore volume between 0.59 and 1.35 cm³/g. The material obtained at 10 K/min is the one that presents the greatest similarity in textural properties to the starting carbon material, differing from the latter only in pore volume, which is bigger for the monolith. It must be noted that both are obtained with the same heating rate. If the xerogel is compared with the other four monoliths, more significant differences are found. It should also be emphasized that these conclusions are valid when water is used as a solvent. Similar studies would be necessary when other solvents are used to see if the trend is the same.

When comparing the monoliths prepared under identical conditions but with diverse carbonaceous sources (Darco vs. CXG: D_$H_2O$ vs. 10 K/min), the differences in textural properties are evident. When the added carbon is xerogel (obtained by a procedure similar to that used for the manufacture of the monolith), the properties of the final monolith do not differ significantly from those of the starting carbon. The contrary is observed for the monoliths prepared with Darco as a carbon source. The latter suggests that the properties of the monolith are greatly influenced by the added carbonaceous source in the first place and by the organic solution and pyrolysis afterward.

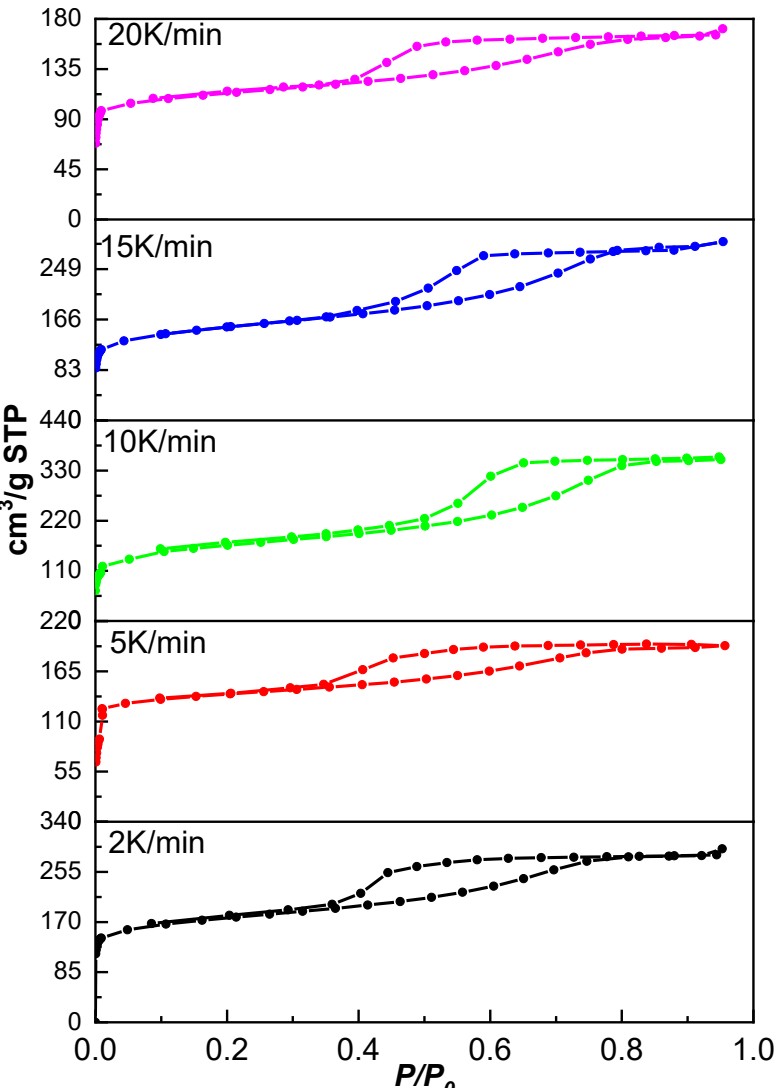

**Figure 6.** $N_2$ adsorption and desorption isotherms at 77 K for the carbonaceous monoliths obtained at different carbonization rates.

Finally, we should emphasize that this method of preparation produces materials with well-defined surface characteristics such as micro and mesoporosity, very attractive characteristics for different catalytic processes that prefer inhomogeneous small/large pore distributions in the same support.

### 2.3. Catalytic Study

Formic acid dehydrogenation reaction has been chosen to test a representative sample of the structured carbon devices. Formic acid has proven to be a good liquid organic hydrogen carrier (LOHC), presenting a safer option for energy storage due to its high volumetric hydrogen capacity (53 g $H_2$/L), low toxicity, and flammability under ambient conditions [27,28]. Moreover, being liquid at room temperature, its handling is comparable to that of diesel and petrol, making it easy to transport and refuel [29]. Another relevant aspect of using formic acid as a hydrogen storage material is that the liberated $CO_2$

after dehydrogenation can be hydrogenated subsequently again to formic acid resulting in a carbon-free process for hydrogen production [30,31].

Formic acid decomposition (FAD) implies two thermodynamically stable reactions: the dehydrogenation reaction (Equation (1)) and the dehydration reaction (Equation (2)).

$$HCOOH \rightarrow H_2 + CO_2 \tag{1}$$

$$HCOOH \rightarrow H_2O + CO \tag{2}$$

Considering the extremely low CO tolerance of the fuel cell (FC), a complete selectivity to formic acid dehydrogenation is desired for all FAD systems [32]. For the catalytic study, palladium has been selected as the active phase, a metal proven to be very effective in this reaction, with better performance than other metals [32,33].

Thus, low charge 0.2%Pd/C monoliths with 9 channels, a length of 29 mm, and a diameter of 17 mm have been prepared (Figure 7), as described in Section 3.3. The manipulation and activation in the HNO₃ acid and palladium deposition process do not alter the physical integrity of the monolith.

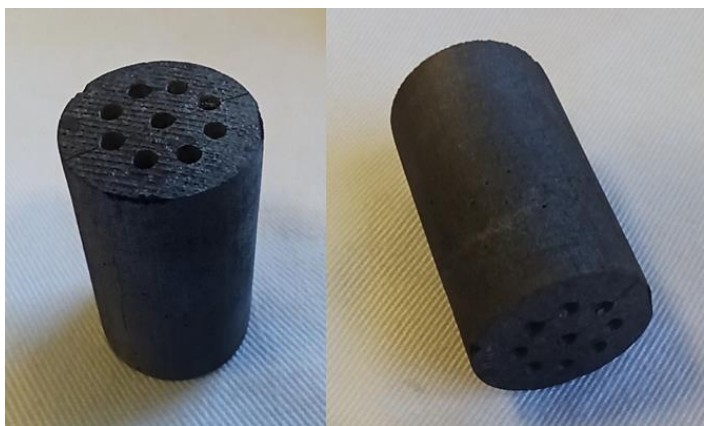

**Figure 7.** Monolith used in the catalytic test.

SEM images and EDX analysis (Figure 8) allow us to corroborate the presence of Pd nanoparticles on the monolith carbon surface with a very homogeneous distribution.

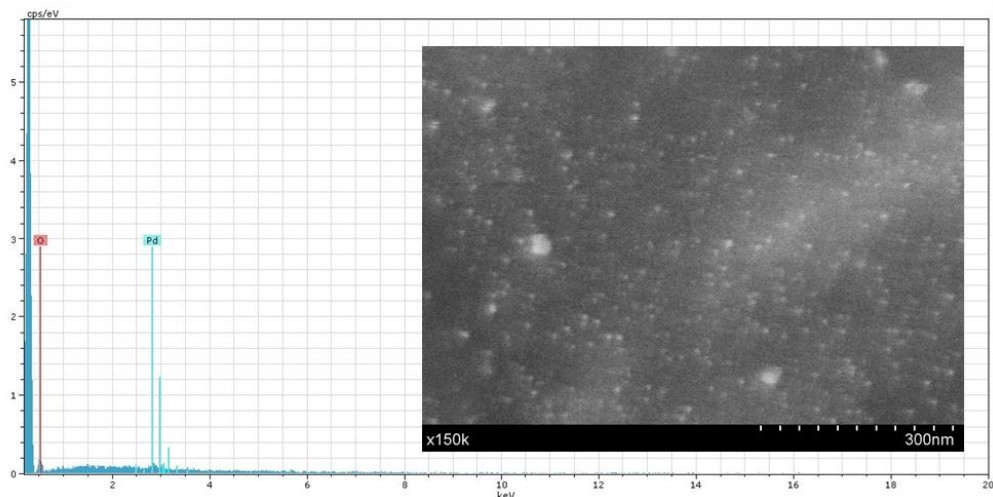

**Figure 8.** SEM micrograph and EDX analysis of the 0.2%Pd/C monolith.

In order to evaluate not only the catalytic activity but also the stability and resistance of our structured system, we have carried out a long-time experience (30 h in a continuous

flow reaction). Figure 9 presents gas-phase FAD activity in terms of $H_2$, $CO_2$, CO, and $CH_4$ flows (mol/min) vs. time at a fixed temperature of 423 K. Because of the high exothermicity of the formic acid dehydrogenation reaction and the low thermal conductivity of the carbon substrate; we decided to test the FAD in a long-term study at a temperature with a high (but not total) formic acid conversion, in order to assure the stabilization of the reaction and to observe catalyst deactivation if it occurs. A previously reported blank experiment with SiC shows that formic acid thermal decomposition starts at 548 K, reaching complete conversion at temperatures above 623 K [32], so the observed conversion in our case is attributable to the catalytic process. The results show that hydrogen production stabilizes after 1300 min of reaction, reaching a constant production of around $1.05 \times 10^{-4}$ mol/min. CO production is five times lower than the hydrogen one, indicating that the dehydrogenation reaction is dominant and $H_2$ is formed selectively. Accordingly, the $H_2$-to-$CO_2$ molar ratio remains close to 1 (1.1) once the steady state is achieved after 200 min of reaction. No $CH_4$ production is detected, confirming the absence of CO or $CO_2$ hydrogenation reactions. No evidence of gasification of carbon substrate is detected in the reaction conditions tested.

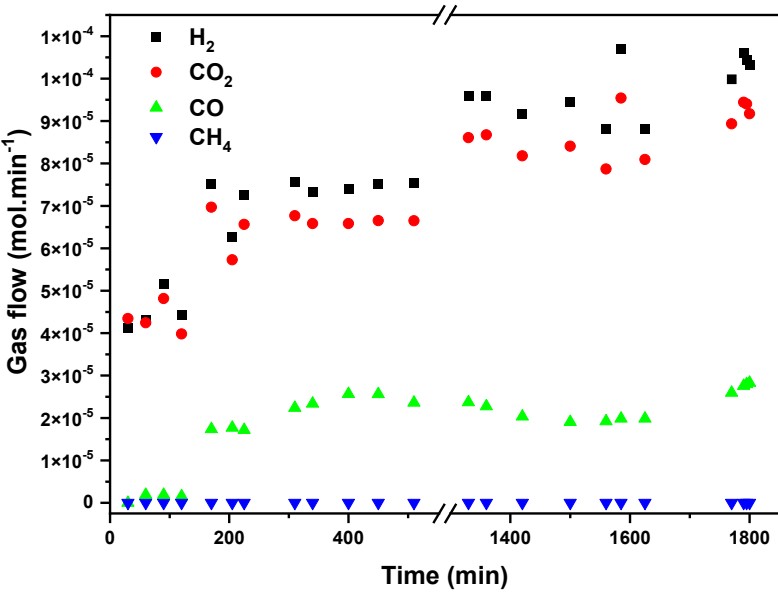

**Figure 9.** Gaseous product flows (mol/min) vs. time at a fixed temperature of 423 K.

The conversion of formic acid is calculated from both generated hydrogen and carbon-containing compounds, giving an average value of 81% for the former and 92% for the latter (Figure 10) once the system stabilizes.

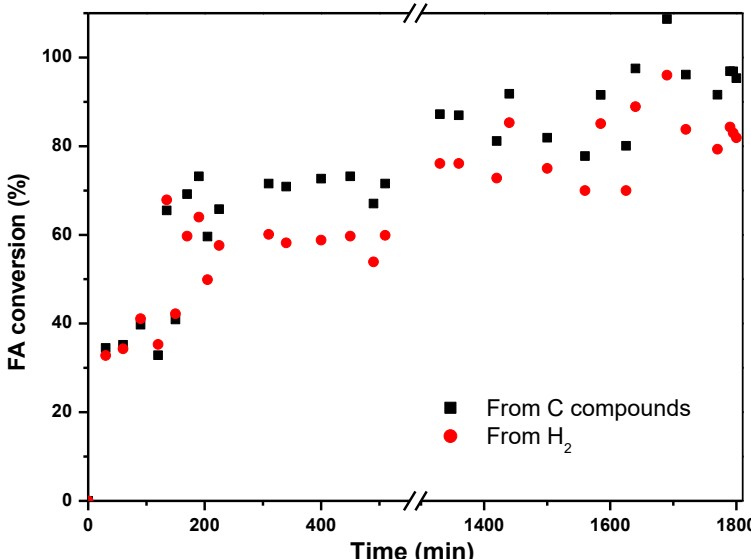

**Figure 10.** Formic acid conversion (%) vs. time at a fixed temperature of 423 K.

The discrepancy between the two values can be attributed to the occurrence of a dehydration reaction that produces CO. It is even more clear when calculating the selectivities for both reactions, where the selectivity for dehydrogenation stabilizes around 80%, while the dehydration remains at 20% (Figure 11). The system shows high selectivity towards dehydrogenation of formic acid at low temperatures (423 K). According to Ruiz-López et al. [32], the presence of water in the system favors dehydrogenation via the Le Chatêlier principle, but also the water–gas shift reaction ($CO + H_2O \rightarrow CO_2 + H_2$) could take place at low temperatures, with the transformation of produced CO to $CO_2$. In fact, Solymosi et al. found that pure $H_2$ cannot be obtained through formic acid decomposition at temperatures above 323 K in the absence of water [34].

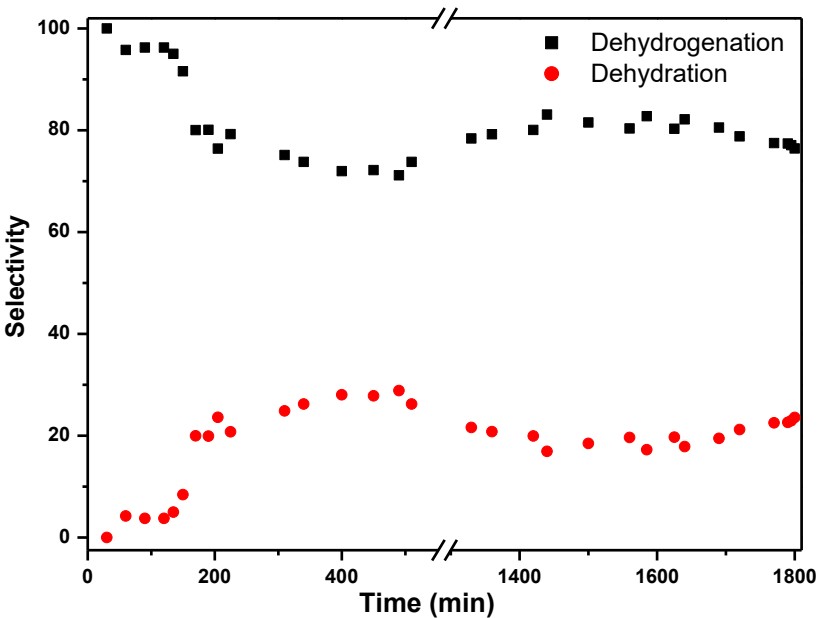

**Figure 11.** Selectivity (%) vs. time at a fixed temperature of 423 K.

The starting shape, characteristics, and weight of the post-reacted monolith remain unaltered after the reaction, indicating that the monolith presents appropriate mechanical

properties and chemical stability for this application. Detailed studies of the catalytic performances of our structured systems will be treated in future works.

### 3. Materials and Methods

Using 3D technology (Software FreeCAD version 0.16 and Ultimaker 3 printer, with incorporated CURA software allowing the importation of FreeCAD projects to generate the printing conditions), molds can be designed and printed in different materials, typically polyvinyl alcohol (PVA), polylactic acid (PLA), or polyurethane (PU). These pieces are employed as negative templates of the final desired shape of carbonaceous structured devices.

The process consists of five steps: the preparation and stabilization of an organic solution (OS) by catalyzed polymerization of resorcinol-formaldehyde mixture; the grinding and sieving of an additional carbonaceous source; the mixing of this carbonaceous source with the organic solution and its packing in a mold obtained by 3D printing; the curing of the polymeric mix and the pyrolysis of the material (Figure 12).

The OS is prepared by dissolving, in the targeted solvent, resorcinol (98% Sigma-Aldrich), sodium carbonate as catalyst ($Na_2CO_3$ anhydrous, Sigma-Aldrich), in a resorcinol-to-carbonate ratio of 300, starch (soluble, ACS Reagent, Sigma-Aldrich) as a binder (between 1–15% by weight with respect to the amount of initial resorcinol) and formaldehyde (Sigma-Aldrich, 37%wt in $H_2O$, 10–15% methanol as stabilizer) in molar ratio 1:2 with respect to resorcinol. The OS is left to stabilize for 24 h at 293 K. The carbonaceous source is ground and sieved to a particle size of less than 600 μm. The charcoal-OS mixture is filled into the mold, covered, and placed in an oven at 333 K for 120 h. The carbon/OS ratio varies according to the used carbon source. The pyrolysis process is carried out in a tube furnace, typically at 1073 K for 2 h with a heating rate of 10 K/min in a $N_2$ flow of 100 mL/min.

Different parameters could be modified in the described procedure, such as solvent nature, presence, type and quantity of catalyst, type and quantity of binder, nature of the additional carbon (commercial activated carbon, charcoal, biochar or nanotubes, among others), or organic solution/carbon ratio. It is also possible to demold the monolith before the pyrolysis step. All these factors affect the characteristics of the finally obtained structured carbon. The conditions of the pyrolysis step (atmosphere, temperature, physical and/or chemical activation, heating rate, time, etc.) also allow the optimization of carbon textural and surface properties.

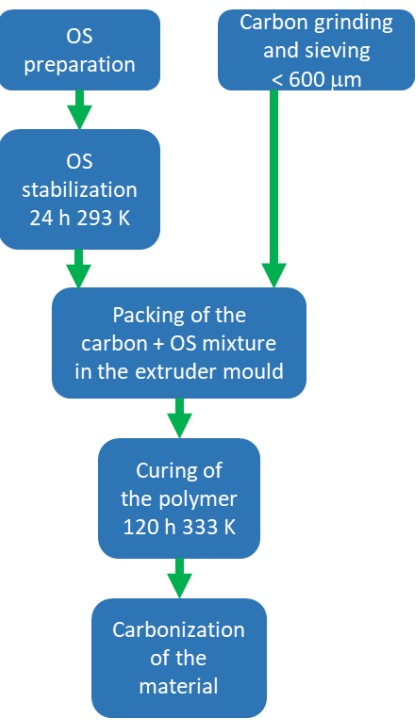

**Figure 12.** Schematic of the manufacturing process for structured carbon devices.

The parameters varied during the study of the influence of the nature of the solvent used in the polymerization stage and the heating rate in the pyrolysis stage on the properties of the final carbonaceous structure are detailed below.

### 3.1. Influence of the Nature of the Solvent in the Polymerization Stage on the Carbon Structured Device

Different solvents have been tested: water, 96% ethanol, polyethylene glycol (PEG 400 PS Panreac), or PEG + 1% polyvinyl alcohol (PVA, 4-88 Mowiol Sigma-Aldrich), maintaining constant the rest of the variables and the additional carbonaceous source (activated charcoal Darco from Sigma-Aldrich). For the preparation of OS, 9.91 g resorcinol is dissolved in 18.8 mL of the solvent, and 13.5 mL of methanol-stabilized formaldehyde (resorcinol-formaldehyde molar ratio 1:2), 0.036 g sodium carbonate (resorcinol/catalyst molar ratio 300), and 0.496 g starch (5% relative to the amount of resorcinol) are added under constant stirring. The mixture is kept stirred until the pH of the solution stabilizes (pH~6.5). The carbonaceous structured devices were obtained following the process described above, fixing a ratio of 0.25 g C/mL for the mixture C/OS.

### 3.2. Influence of the Heating Rate in the Pyrolysis Step on the Carbon Structured Device

Different pyrolysis heating rates were studied by preparing massive monoliths following the methodology previously exposed, using water as a solvent for the organic solution and a carbon xerogel (CXG) as the carbonaceous source. The heating rate range used was between 2 and 20 K/min, maintaining constant the rest of the variables.

To obtain the carbon xerogel, 9.910 g resorcinol, 0.496 g starch (5% by weight of resorcinol), and 0.036 g sodium carbonate (resorcinol/catalyst molar ratio 300) were dissolved in 18.8 mL water under continuous stirring. After 20 min, the pH of the solution stabilized (~6.7), and 13.5 mL of methanol-stabilized formaldehyde was added. The mixture was poured into closed cylindrical glass vials and kept for 24 h at room temperature and 120 h at 333 K. After the gelation step, a dark red solid was obtained and pyrolyzed under nitrogen flow in a tube furnace (100 mL/min). The heating program included a ramp up

of 5 K/min to 473 K, maintained for 30 min, and a second ramp of 10 K/min to 1073 K, maintained for 3 h [35].

### 3.3. Catalytic Study

For the catalytic study, 0.2%Pd/C structured devices were prepared. Monoliths with 9 channels, a length of 29 mm, and a diameter of 17 mm have been obtained using PLA molds, water as a solvent for OS preparation, and a pyrolysis heating rate of 10 K/min.

The carbon used for the manufacture of these monoliths was obtained from pyrolyzing crystalline microcellulose. The pyrolysis conditions were as follows: $CO_2$ flow of 200 mL/min, RT heating up to 963 K at 10 K/min, kept at that temperature for 5 min; heating at 1 K/min up to 1023 K, kept at that temperature for 60 min.

Before palladium deposition, the monolith is activated with $HNO_3$ in order to increase its hydrophilicity. For this purpose, it is immersed for 24 h in concentrated $HNO_3$ (Panreac Química S.A., 69%) and then washed with deionized water until reaching the neutral pH of the water and dried at 373 K.

The chemical precursor for Pd was palladium acetate (Johnson Matthey PLC, 47.15%). To deposit the palladium, an adequate quantity of Pd precursor was dissolved in acetone (Sigma, 90%), and the carbon monolith, pre-heated at 373 K, was submerged into the solution. Then, it is dried at 373 K, and the process is repeated until no Pd solution remains. Prior activity measurements, the catalysts were treated thermally at 523 K for 1 h in an inert atmosphere ($N_2$, 100 mL/min) and then reduced at 573 K for 1 h ($N_2/H_2$, 1:1, total flow = 100 mL/min).

### 3.4. Characterization

X-ray diffraction (XRD) spectra were recorded using an X'Pert Pro PANalytical instrument working with Cu-$K_\alpha$ (40 mA, 45 kV) with 0.05° step size and 300 s of step time over a 2θ range from 10 to 80°. The structural parameters of the carbonaceous materials, such as inter-layer spacing ($d_{002}$), crystallite height (Lc), and crystallite diameter (La), were determined using the Braggs (Equation (3)) and Scherrer equations (Equations (4) and (5)).

$$d_{002} = \frac{\lambda}{2\sin\theta_{002}} \tag{3}$$

$$L_c = \frac{K_c\lambda}{\beta_{002}\cos\theta_{002}} \tag{4}$$

$$L_a = \frac{K_a\lambda}{\beta_{100}\cos\theta_{100}} \tag{5}$$

where $\lambda$ is the wavelength of the incident X-ray, Cu-$K_\alpha$ 1.5405 Å; $\theta_{002}$ and $\theta_{100}$ are the peak position of (002) and (100) planes in degrees; $\beta_{002}$ and $\beta_{100}$ are the full width at half maximum (FWHM) of the corresponding diffraction peaks; Kc is 0.89 and Ka 1.84 [36,37].

Textural properties were studied by $N_2$ adsorption-desorption at 77 K in a +Micromeritics Tristar II instrument 2000. Before each analysis, the samples were degassed at 523 K for 12 h under a vacuum. The BET equation and the t-plot method were used to calculate specific area, micropore area, and volume, respectively. The density functional theory for determining the predominant average pore diameter ($APD_{DFT}$) in carbonaceous materials was used.

Scanning electron microscopy images were obtained on a HITACHI S-4800 SEM-FEG equipped with secondary and backscattered electron detectors. A voltage of 2 kV was used to obtain the SEM micrographs, and a voltage of 20 kV to obtain the EDX analyses.

For the catalytic study, a stainless-steel reactor (250 mm in length, 1.7 mm in internal diameter) was used, fed with a pre-heated inlet stream, using a syringe pump, an evaporator, and a mixer to homogenize the reaction flow. A heat exchanger was used to condense the outlet liquid phase (water and non-reacted formic acid), and the gas phase ($H_2$,

CO, $CO_2$, and $CH_4$) was continuously monitored using an ABB AO2020 analyzer. A 100 mL·min$^{-1}$ (5% *v/v* formic acid, 25% *v/v* distilled water, and 70% *v/v* $N_2$) flow fed the reactor, and the gas hourly space velocity (GHSV) was about 910 h$^{-1}$. At the selected temperature of 423 K, a long-term experiment was performed for 30 h to test the catalyst's stability. The formic acid conversion and the selectivities to dehydrogenation and to dehydration were calculated following Equations (6)–(9):

$$\text{Formic acid conversion (from C compounds)}(\%) = \frac{n_{CO_2} + n_{CO} + n_{CH_4}}{n_{FA}^0} \cdot 100 \tag{6}$$

$$\text{Formic acid conversion (from H}_2)(\%) = \frac{n_{H_2}}{n_{FA}^0} \cdot 100 \tag{7}$$

$$\text{Selectivity to dehydrogenation, }(\%) = \frac{n_{CO_2}}{n_{CO_2} + n_{CO}} \cdot 100 \tag{8}$$

$$\text{Selectivity to dehydration, }(\%) = \frac{n_{CO}}{n_{CO_2} + n_{CO}} \cdot 100 \tag{9}$$

where $n_{FA}^0$ is the FA molar flow fed to the reactor and $n_i$ is the obtained molar flow for the corresponding species. The formic acid conversion was also checked by HPLC recovering condensate at each temperature after the reactor (column Hi-Plex H, milliQ water as mobile phase).

## 4. Conclusions

The proposed methodology makes the fabrication of integral carbon monoliths possible using a resorcinol-formaldehyde polymer resin containing starch. The incorporation of this low-cost binder improves the structural properties of the monoliths, generates controlled microporosity, and increases the phenolic ring crosslinking upon reaction with the synthetic polymer.

The control of the process parameters allows the manufacture of carbon structures with different three-dimensional geometries and the number of cells per unit area, as well as different textural and surface properties, thus adapting its characteristics to be used as a support or catalyst in different catalytic reactions or any other application of interest.

The present work also explores the influence of the solvent nature in the polymerization stage, the carbonaceous source, and the heating rate during pyrolysis on the characteristics of the obtained carbonaceous device. Using different solvents is possible, giving the advantage of working with hydrophilic, hydrophobic, or amphoteric carbons and expanding the field of carbon devices application. The monolith's textural properties are influenced by the added carbonaceous source and the carbonization process of the organic solution.

The presence of combined micro- and mesoporosity in the structures is attractive for catalytic processes where a heterogeneous pore distribution is preferred, with pores both smaller or larger than 2 nm on the same support.

The proposed strategy allows obtaining 100% carbon structures, but it could also be extrapolated to obtain structures of metal oxides, ceramics, hybrid inorganic/inorganic materials, etc.

The prepared carbon-structured devices also prove to be suitable as supports for palladium-based catalysts, achieving high conversions and selectivity toward hydrogen formation under formic acid dehydrogenation at low temperatures (423 K).

**Author Contributions:** Conceptualization, S.I. and Y.Y.A.; methodology, N.R., G.D.-M., M.M.T., E.R.-L., S.I., and M.I.D.; validation, S.I. and M.Á.C.; formal analysis, M.I.D.; investigation, N.R., G.D.-M. and E.R.-L.; data curation, M.M.T. and M.I.D.; writing—original draft preparation, G.D.-M.

and M.I.D.; writing—review and editing, N.R., M.M.T., Y.Y.A., E.R.-L., M.Á.C., and M.I.D.; supervision, Y.Y.A. and M.Á.C.; project administration, M.Á.C.; funding acquisition, M.Á.C. and S.I. All authors have read and agreed to the published version of the manuscript.

**Funding:** This research was funded by the Ministerio de Ciencia e Innovación (MCIN/AEI/10.13039/501100011033/) grant number [ENE2017-82451-C3-3-R and PID2020-113809RB-C32] and Junta de Andalucía via Consejería de Transformación Económica, Industria, Conocimiento y Universidades, grant number [P18-RT-3405] co-financed by FEDER funds from the European Union.

**Data Availability Statement:** Data are available upon request.

**Acknowledgments:** E. Ruiz-López would like to acknowledge the Spanish Ministerio de Universidades and the Unión Europea-NextGeneration EU for the financial support of her Margarita Salas Fellowship.

**Conflicts of Interest:** The authors declare no conflict of interest.

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
