# Peer review of "New 3D Printing Strategy for Structured Carbon Devices Fabrication"

_catalysts, doi:10.3390/catal13071039_

Round 1

Reviewer 1 Report

Greetings, Editor thank you for providing me with the opportunity to review the article. I reviewed the article with Carbon microreactors prepared by a new 3D-based printing strategyThe article topic is intriguing and promising in the area. Overall, the article structure and content are suitable for journal standard. I am pleased to send some comments which need to be corrected before publication. Please consider these suggestions as listed below.

The language of the article should be improved.

Please make the title more catchy. (not just a sample sentence)

Please add numerical results in abstract and revise it (not like introduction)

Need to write the novelty of this work in introduction section.

Need to replace blur figures with high quality. and make sure that all mentioned in the text on proper space.

Need to rewrite the introduction in a  scientific flow ..like problem statement, solution etc.

There are some typo, space and spelling errors that should be rectified.(such as the full stop in case of captions of each figure, font size of the figure 6 etc )

Please make uniform the format of the whole manuscript.

Please rewrite the conclusion in concise way.

The language of the article should be improved.

Author Response

Response to reviewer’ comments

The authors acknowledge the reviewer’ comments and provide below detailed responses and description of the changes included in the manuscript.

In the revised manuscript, English has been improved and the introduction section modified, displacing paragraphs of the tested catalytic reaction to the results (2.3) section. Accordingly, some references have been renumbered.

On the other hand, the title of the work has been changed, figures 1,4 and 6 have been replaced by others of higher quality, and figure 8 has been modified to include EDX analysis.

Other changes have been highlighted in yellow.

Referee 1

Comments and Suggestions for Authors

Greetings, Editor thank you for providing me with the opportunity to review the article. I reviewed the article with Carbon microreactors prepared by a new 3D-based printing strategy.  The article topic is intriguing and promising in the area. Overall, the article structure and content are suitable for journal standard. I am pleased to send some comments which need to be corrected before publication. Please consider these suggestions as listed below.

  1. The language of the article should be improved.

We have revised the manuscript and improved the English as requested.

  1. Please make the title more catchy. (not just a sample sentence)

According to your suggestion, we have changed the tittle. Now, the manuscript it is entitled “New 3D printing strategy for structured carbon devices fabrication”

  1. Please add numerical results in abstract and revise it (not like introduction).

Thanks for your suggestion.

The article focuses mainly on the benefits of our proposed method for the preparation of carbonaceous structured systems rather than a specific application. So, numerical results are not so evident to be incorporated in the abstract. In any case, following the reviewer's indications, the abstract has been modified and numerical data related to the conversion and selectivity achieved in the catalytic example presented have been included.

  1. Need to write the novelty of this work in introduction section.

Many thanks for your suggestion.

We have modified the introduction section, in order to reinforce the novelty of our work. Additionally, we have displaced the paragraphs of the tested catalytic reaction (formic acid dehydrogenation) from the introduction to the results (2.3) section.

  1. Need to replace blur figures with high quality. and make sure that all mentioned in the text on proper space.

In accordance with your suggestions, figures 1 and 4 have been replaced by figures of higher quality and all figures have been placed in the vicinity of the text where they are referred to.

  1. Need to rewrite the introduction in a scientific flow ..like problem statement, solution etc.

Many thanks for your suggestion. We have modified the introduction as requested.

  1. There are some typo, space and spelling errors that should be rectified (such as the full stop in case of captions of each figure, font size of the figure 6 etc ).

Thank you very much for your careful reading of our manuscript. Detected typographical errors have been rectified, including font size in figure 6. In the case of the full stop in the figure captions, this is the format of the publication.

  1. Please make uniform the format of the whole manuscript.

We have checked the whole manuscript and tried to uniform it.

  1. Please rewrite the conclusion in concise way.

As suggested, the conclusions have been rewritten to make them more concise.

Comments on the Quality of English Language

The language of the article should be improved.

Done, as requested.

Reviewer 2 Report

In this manuscript, the authors present an efficient approach to fabricate pure carbon structured catalyst devices with tailorable textural and surface properties by deliberately applying different synthesis parameters. Combining X-ray diffraction microstructural analysis, SEM surface morphology analysis, N2 adsorption and desorption and long-term gas-phase FAD study, it is demonstrated that the developed carbon structured device is an effective palladium support for gas-phase formic acid dehydrogenation reaction. The results from this work show the potential of using 3-D printing based approach in developing carbon-based catalyst device with outstanding performance. The presented results seem reasonable and interesting; therefore, I suggest the publication of this manuscript in Catalyst, with following comments.

1)     I suggest the author moves section 3 focusing on materials and methods to be in front of the section 2.

2)     I suggest the author includes the EDX results mentioned on page 11 in the main manuscript.

3)     For studying the effect of pyrolysis heating rates on the structures of the obtained carbon device, the author used water as solvent. Will applying other types of solvent influence the effect of heating rate on structures and performance of the carbon structured device? I suggest the author to talk about this in the current manuscript.

Some minor English language editing are required.

Author Response

Response to reviewer’ comments

The authors acknowledge the reviewer’ comments and provide below detailed responses and description of the changes included in the manuscript.

In the revised manuscript, English has been improved and the introduction section modified, displacing paragraphs of the tested catalytic reaction to the results (2.3) section. Accordingly, some references have been renumbered.

On the other hand, the title of the work has been changed, figures 1,4 and 6 have been replaced by others of higher quality, and figure 8 has been modified to include EDX analysis.

Other changes have been highlighted in yellow.

Referee 2

Comments and Suggestions for Authors

In this manuscript, the authors present an efficient approach to fabricate pure carbon structured catalyst devices with tailorable textural and surface properties by deliberately applying different synthesis parameters. Combining X-ray diffraction microstructural analysis, SEM surface morphology analysis, N2 adsorption and desorption and long-term gas-phase FAD study, it is demonstrated that the developed carbon structured device is an effective palladium support for gas-phase formic acid dehydrogenation reaction. The results from this work show the potential of using 3-D printing based approach in developing carbon-based catalyst device with outstanding performance. The presented results seem reasonable and interesting; therefore, I suggest the publication of this manuscript in Catalyst, with following comments.

  1. I suggest the author moves section 3 focusing on materials and methods to be in front of the section 2.

I agree with you it could be more logical to place Materials and Methods section before Results and Discussion section, but the used order is according to MDPI format.

  1. I suggest the author includes the EDX results mentioned on page 11 in the main manuscript.

EDX result has been included in figure 8, as suggested.

  1. For studying the effect of pyrolysis heating rates on the structures of the obtained carbon device, the author used water as solvent. Will applying other types of solvent influence the effect of heating rate on structures and performance of the carbon structured device? I suggest the author to talk about this in the current manuscript.

Of course, the use of other solvents than water may lead to different results. As we have tried to explain in the whole manuscript, our proposed method has a lot of synthetic parameters than can be modified to some extent, allowing to prepare a large variety of carbonaceous structures with different properties and performances. The nature of the solvent, is one of them and should be studied in the near future.

A comment has been included in the manuscript on page 9.

Comments on the Quality of English Language

Some minor English language editing are required.

Done, as requested.

Reviewer 3 Report

The study presents a methodology to produce carbon monoliths by incorporating a low-cost binder, starch, into a resin. This improves the structural properties and microporosity of the monoliths. Various parameters can be controlled to obtain carbon structures with different geometries and properties. The choice of solvent and carbonaceous source influences the textural properties of the monoliths. The monoliths exhibit micro- and mesoporosity, making them suitable for catalytic processes. The methodology can be extended to other materials. The carbon monoliths serve as effective catalyst supports for hydrogen production from formic acid at low temperatures.

On page 6 Figure 4: How do the observed differences in the isotherms and hysteresis cycles between the commercial activated carbon and the monoliths impact their adsorption properties and suitability for different applications? Are there any specific advantages or disadvantages associated with the presence of mesopores, micro and mesopores, or graphite structures in the carbon materials?

On page 12 of the manuscript. How reliable is the chosen temperature of 423 K for the long-term study? Are there any potential drawbacks or limitations associated with this temperature selection?

On page 10 of the manuscript: What are the implications of the observed variations in textural properties, such as surface area, pore volume, and porosity, between the monoliths obtained at different heating rates? How do these variations affect the overall performance or applications of the monoliths?

Moderate editing of English language required

Author Response

Response to reviewer’ comments

The authors acknowledge the reviewer’ comments and provide below detailed responses and description of the changes included in the manuscript.

In the revised manuscript, English has been improved and the introduction section modified, displacing paragraphs of the tested catalytic reaction to the results (2.3) section. Accordingly, some references have been renumbered.

On the other hand, the title of the work has been changed, figures 1,4 and 6 have been replaced by others of higher quality, and figure 8 has been modified to include EDX analysis.

Other changes have been highlighted in yellow.

Referee 3

Comments and Suggestions for Authors

The study presents a methodology to produce carbon monoliths by incorporating a low-cost binder, starch, into a resin. This improves the structural properties and microporosity of the monoliths. Various parameters can be controlled to obtain carbon structures with different geometries and properties. The choice of solvent and carbonaceous source influences the textural properties of the monoliths. The monoliths exhibit micro- and mesoporosity, making them suitable for catalytic processes. The methodology can be extended to other materials. The carbon monoliths serve as effective catalyst supports for hydrogen production from formic acid at low temperatures.

  1. On page 6 Figure 4: How do the observed differences in the isotherms and hysteresis cycles between the commercial activated carbon and the monoliths impact their adsorption properties and suitability for different applications? Are there any specific advantages or disadvantages associated with the presence of mesopores, micro and mesopores, or graphite structures in the carbon materials?

The possible applications of a carbon depend to a large extent on its textural properties. For example, highly microporous carbons with high specific surface areas are used as adsorbents, while carbons with a higher percentage of mesoporosity are required for catalytic applications. In this case, according to the shape of the isotherms and their quantification, the largest contribution to the total area (BET) is given by both micropores and smaller mesopores. Depending on the solvent, the decrease in micropores is compensated by an increase in mesopores. Thus, each type of application requires different textural properties of the carbonaceous support, and what may be an advantage for one application may be a disadvantage in another. In this sense, the aim of this work is to demonstrate that the proposed method allows modulating these characteristics, through the choice of the carbonaceous source, the solvent or the modification of the pyrolysis conditions. However, in the examples shown in the manuscript, the textural properties of the prepared monoliths are not so different as to be used in very diverse applications, being all of them suitable for catalytic purposes. Therefore, we have selected an example of an application based on heterogeneous catalytic reactions.

  1. On page 12 of the manuscript. How reliable is the chosen temperature of 423 K for the long-term study? Are there any potential drawbacks or limitations associated with this temperature selection?

As stated in the manuscript, this temperature was chosen because it is below the temperature at which the thermal decomposition of formic begins, so that the observed conversion is attributable to the catalytic process. Moreover, at the selected temperature, the conversion is high but below 100% in order to clearly observe catalyst deactivation if it occurs. If mechanistic studies were to be carried out, it would be necessary to work at temperatures where the conversions were lower than 20%, but this type of study is outside the scope of this work.

The discussion has been improved accordingly on page 11.

  1. On page 10 of the manuscript: What are the implications of the observed variations in textural properties, such as surface area, pore volume, and porosity, between the monoliths obtained at different heating rates? How do these variations affect the overall performance or applications of the monoliths?

Please, take advice of our response to your point 1.

The use of different heating rates influences the textural properties of the final carbon. The high possibility of designing the texture in these materials, varying the heating rate without modifying the shape of the isotherm or the hysteresis is interesting because it gives the process versatility. In addition, the presence of mesopores in a microporous structure enhances the diffusion of the reagents towards the active sites housed in the micropores. So, in principle, the adequate selection of the parameters of synthesis could drive to important changes in their textural characteristic. However, in the examples shown in the manuscript, the textural properties of the prepared monoliths are not so different to a be used in very different applications.

Comments on the Quality of English Language

Moderate editing of English language required

Done, as requested.

Round 2

Reviewer 3 Report

The authors have addressed the concerns raised by the referees.